# AUTONOMOUS VEHICLE FLEET COORDINATION WITH DEEP REINFORCEMENT LEARNING

## ABSTRACT

Autonomous vehicles are becoming more common in city transportation. Companies will begin to find a need to teach these vehicles smart city fleet coordination. Currently, simulation based modeling along with hand coded rules dictate the decision making of these autonomous vehicles. We believe that complex intelligent behavior can be learned by these agents through Reinforcement Learning. In this paper, we discuss our work for solving this system by adapting the Deep Q-Learning (DQN) model to the multi-agent setting. Our approach applies deep reinforcement learning by combining convolutional neural networks with DQN to teach agents to fulfill customer demand in an environment that is partially observable to them. We also demonstrate how to utilize transfer learning to teach agents to balance multiple objectives such as navigating to a charging station when its energy level is low. The two evaluations presented show that our solution has shown that we are successfully able to teach agents cooperation policies while balancing multiple objectives.

## 1    INTRODUCTION

Many business problems that exist in todays environment consist of multiple decisions makers either collaborating or competing towards a particular goal. In this work, the challenge is applying multi-agent systems for autonomous fleet control. As Autonomous Vehicles (AVs) are becoming more prevalent, companies controlling these fleets such as Uber/Lyft will need to teach these agents to make optimal decisions. The goal of this work is to train these agents/cars optimal relocation strategies that will maximize the efficiency of the fleet while satisfying customer trip demand. Traditional solutions will use discrete event simulation modeling to optimize over a chosen objective function. This approach requires various hand coded rules as well as assumptions to help the model converge on a solution. This becomes an extremely difficult problem when there are many outside environment dynamics that can influence an agents/cars decision making (E.g. Charging, Parking). Furthermore, a solution to a particular environment may become outdated with new incoming information (E.g. New Demand Distribution).

An algorithm that can adapt and learn decision making organically is needed for these types of problems and recent works in Reinforcement Learning and particularly Deep Reinforcement Learning has shown to be effective in this space. Deep Minds recent success with Deep Q Learning (DQN) was proven to be very successful in learning human level performance for many Atari 2600 games which was difficult before this because of its highly dimension unstructured data. In this work, we will pull from prior work in Multi-Agent Deep Reinforcement Learning (MA-DRL) and extend this to our multi-agent system of cars and fleet coordination. We will represent the city environment that holds the cars and customers as an image-like state representation where each layer holds specific information about the environment. We then will introduce our work with applying this to a partially observable environment where agents can only see a certain distance from them and show how this helps with scaling up. Along with that, we will show how we took advantage of Transfer Learning to teach agents multiple objects in particular charging an import aspect of AVs. Our results show that we are successfully able to teach coordination strategies with other cars so that they can optimize the utility of each car. Finally, we are also able to teach agents the second object of keeping itself alive while not losing the previous objective of picking up customers.

## 2 RELATED WORKS

The domain of Deep Reinforcement Learning has garnered increased attention recently because of its effectiveness in solving highly dimensional problem spaces. Although its sub-field of multi-agent systems presents a difficult challenge in how we represents other agents in the environment. Since these agents are non-stationary, how do we train agents to intelligently cooperate and find an optimal policy when each agent's rewards depend on one another? [Tampuu, Ardi] builds on DeepMind success with the Atari game pong and tests cooperation and competitive policies with each player. [Foerster, Jakob] proposes two novel approaches (RIAL,DIAL) which allows error to backpropagate through multiple agents. Although, this approach has only been tested on riddles where there is a small number of agents and does not seem to scale well to larger problems. [Palmer, Gregory] solves the experience replay memory becoming outdated problem of multi-agent system. [Egorov, Maxim] introduces a novel approach to solve multi-agent systems through convolutional neural networks, stochastic policies, residual networks and is what we will building our solution for the large scale autonomous fleet control on. We will extend some of this work to partially observable and multi-objective case of our agents.

## 3 PROBLEM FORMULATION

### 3.1 ENVIRONMENT OBJECTS

The first step in training a Reinforcement Learning model to to create an environment that is representative of the real world characteristics of your problem. In our case, we needed to create an environment that represents the city dynamics that are associated with the Uber/Lyft ride sharing platform. The general goal of each agent is to travel to customers on the map and fulfill that demand. The different objects and locations that are present in this environment are as follows and can be referenced in the example figure 1.

**Car (Blue Circle):** This is the agent or decision maker that will be acting in the environment. Its actions are right, left, up, down, and stay. The car also has an associated energy level that decrements every timestep until it runs out of batter. Agents can occupy the same space as other agents without collision. Agents can not travel outside of the defined map or into obstacles on the map.

**Customer (Red Circle):** The goal location of where the Agents need to travel to. Once an agents location matched a customers location, then the customer will be considered fulfilled. Customers also have a drop-off location and travel time associated with them that determines how long an agent/car is removed from the system while in transit. Each customer begins with a random wait time of 5-15 timesteps which decrements every step. Once the wait time has been expired the customer is expire and be removed from the map. Finally, customers are generated from a distribution of demand data that we created based on real trip data for the test city location. Each timestep, new customers are generated by pulling from this distribution.

**Obstacles (Gray Square):** These are the locations on the map that the agents cannot travel to. Customer also will not appear at these locations.

**Charging Stations (Yellow Square):** These are the locations that agents can travel to refill their energy levels. Agents must choose the Stay action at these locations for  of their battery to be refilled. Customers can appear at these locations.

**Open Road (Green Square):** These are the standard spaces that agents and customers can exist on.

### 3.2 ENVIRONMENT REWARD STRUCTURE

Each timestep the order of activity are as follows, advance each agent sequentially, resolve their pickups or collisions, decremented car energy levels and customer wait times, remove customers that have waited too long or have been picked up, and finally generate new customers from the distribution. Each one of these transitions will have a reward associated with each agent. These events and and the reward structure are shown in Table 1.

After each timestep the rewards are aggregated and attributed to that specific agent for that initial state. To point out, you may see there is a small negative reward for standard movement. This was

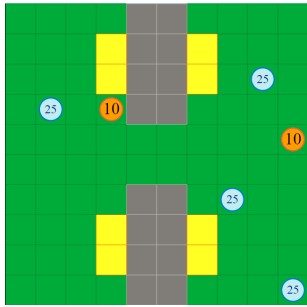

Figure 1: Example environment with agents and customers

Table 1: Event Reward Structure

| Event | Reward |
|---|---|
| Agent picks up customer | +1 |
| Agent collides with obstacle or travels outside the map | -1 |
| A customers wait time runs out | -1 to all agents in vicinity |
| Standard Movement (E.g. left, right) | -0.1 |
| Agent chooses Stay in charging location (Charging) | +0.1 |
| Agent chooses Stay in open road (Parking) | -0.05 |
| Agents energy reaches 0 | -10 |

essential to incentive agents to find the quickest path to the customers. Another important reward structure decision was the small positive reward for charging. We found that there was not enough signal to just have a large negative penalty for losing all of its energy. We had to incentivize being in that charging station space without giving a strong enough reward to detract from the agents picking up the customers.

## 3.3 ENVIRONMENT STATE REPRESENTATION

Now that we defined our various objects and reward structure, we need to represent the environment as a vector that the Deep Reinforcement Learning model can learn from. Along with the suggestions of previous work, we found that the best way to do this while keeping spatial representations was with an image-like structure. Just like how images provide 3 matrices stacked on top of each other to represent *rgb* values, the same can be done for our environment. Our state representation has 5 layers that encode a different piece of information and result in a tensor of *5 X W x H*. The following are what each channels represents.

**Self Layer:** Simply encodes the location of the agent of interest. A value of 1 is given to the (x,y) location where the agent exists. Each self layer is unique to each agent.

**Other Agents Layer:** This encodes the locations of all the other agents excluding yourself. A value of 1 is given to locations where other agents exists and if there are two agents at a location the value 1 is still used.

**Customer Layer:** This encodes the locations of all the customers on the map or within each agent's vision. Here an integer value that represents that customers wait time remaining is placed at the location for that specific customer. When customers are removed from the map than the value will return to 0.

**Obstacles Layer:** This encodes the locations of obstacles and charging locations. The value 1 is used to represent locations where the agent cant go to and the value 2 is used to encode the location of the charging station.

**Extra Agent Information:** This encodes the energy and priority of the agent of interest. For energy we placed the integer value of how much energy is left on the (0,0) location of the matrix and the priority on the (0,1) location

As we will mention later, in the partially observable version of this environment, we limit this state space representation to just be of that specific agents vision distance v. Figure 3, is the visual matrix of how the example map presented above translates to these layers.

## 4 METHODS

In the following sections we will build on the simple Deep Reinforcement Learning technique Deep Q Learning (DQN) and walk through the adaptations that make it the final partially observable multi Agent System (PO-MA-DRL). We will describe how these methods relate to our environment and how they affected our implementation decisions for this environment.

### 4.1 DQN W/ CONVOLUTIONAL NEURAL NETWORKS

The goal of reinforcement learning is to learn an optimal decision making policy that will maximize the future expected reward. One of the most common Reinforcement Learning algorithms is Q-Learning which represents the maximum discounted reward when we perform action a in state s, and continue optimally from that point on. Q-Learning is a form of model-free learning, meaning that it does not have to learn a model of the environment to optimally settle on a policy. The following formula is the Q-Function which is recursively calculated with transitions of *(s,a,r,s)*. In our case, s represents the image like array of the state representation holding all the agents, customers, and obstacles. The values *a* and *r* are the straightforward action taken by the agent and the associate reward. *s* represents the image-like state after the agent has moved to its next location and collision detection or customer pick up has been resolved.

$$Q(s,a) = r + y * maxQ(s',a')  \tag{1}$$

This formula is essentially the Bellman equation which is simply the current reward plus the discounted future reward of the next time step if you were to take the best action. This can be implemented as a large table of states by actions and can be recursively solved if the environment is explored enough. If iterated enough times, the Q-Function should approach the true value of taking a certain action in a state and an optimal decision policy can be looked up by taking the action that will return that maximum value.

When environments become complex with very large state representations, like Atari games, then it becomes computationally difficult to converge on a reasonable Q-Function because the look-up tables can become very large. This is where the recent success in Deep Q Learning has come into play. Neural networks do a great job in generalizing highly dimensional data in a lower space. We can essentially represent that large Q-Function table with a neural network. The Q(a,s) value can be estimated with the net and the loss of this network simply is as follows.

$$L = 1/2 * [r + maxQ(s',a') - Q(s,a)]^2  \tag{2}$$

Along the lines of [Egorov, Maxim], we also decided to use Convolutional Neural Networks (CNNs) to interpret these image like state representations. CNNs have shown to be successful for image classification tasks because of its ability to understand spatial relationships within the image. The same can be said with the importance of the geospatial location of agents in relation to other agents and customers. In addition, CNN's allow us to scale our environment to large sizes without the exponential increase in processing time that may have occurred with fully connected networks.

Finally what makes the convergence of this deep Q-Learning algorithm possible is the addition of Experience Replay, e-greedy exploration, and target networks. Experience Replay allows us to store all of our ¡s,a,r,s¿ transitions in a bucket that we can access. During training, mini batches of transitions are pulled from the bucket and used to fit on the network. E-greedy exploration, allows us to initially randomly take actions but over time the agent will take actions it know result in the maximum expected reward. Lastly, target networks, is a separate network that is a mere copy of the

previous network, but frozen in time. This network is updated at slower intervals than the actual network as to stabilize the convergence of the Q-values during training.

## 4.2 MULTI AGENT DEEP REINFORCEMENT LEARNING (MA-DRL)

The biggest challenge with multi-agent systems is how we are going to optimize one's own control policy while understanding the intentions of the other agents. Referring back to our image representation of the state, we have one layer that represents yourself and another layer that represents the other agents. Now when we defined s as the updated image-like state after the agent has moved earlier, we must now modify that to account for the other agents in the system. The final value s represents the image-like state after all the other agents have moved. So, each agent is not looking at the other agents as a stationary object, rather as intentions that need to be accounted for when making its own optimal decision. When we increment the state from s to s, we must perform all the conflict resolutions of agent-agent and agent-customer interactions before we can resolve the final state. All the agents move sequentially rather than simultaneously which means that agents with the higher priority will be first to pick up customers. We decided to have all the agents controlled by a single network as a sort of fleet manage rather than have each agent train its own network. While we lose the ability to teach agents different personality and pickup styles, for the purpose of scaling up to a large number of agents this was not feasible.

Another incentivization used to create signals for agents to act efficiently with one another is the penalty for missing a customer. Each agent feels the negative reward of missing a customer even if it is on the other side of the map. This incentivises the agents to learn a type of divide and conquer technique as to minimize the probability of missing a future customer. In the Results/Tests section we will show how effective this has been in a simple bridge example. A final consideration was how do we update the replay memory with transitions from multiple agents. We decided to increase the Replay size proportional to the number of agents so that the refresh rate of the memory would be consistent to if it were a single agent. As proposed by [Palmer, Gregory], there can be more efficient ways to update the experience replay for multi-agent systems but we leave that for later work.

## 4.3 PARTIALLY OBSERVABLE MULTI AGENT DEEP REINFORCEMENT LEARNING (PO-MA-DRL)

Partial observability is the final addition to our methods and is where our work differs from some of the related works. When attempted to train our MA-DRL with actual data from a pilot city of Autonomous Vehicles, the problem just became too large with over 200+ cars and a 25x25 grid size. Using CNNs were not enough to scale the processing of the image-like representations. To solve this, we made the agents partially observable to the entire environment or map. We limited how far an agent can see around them as v (e.g. +10 spaces away) which limits how big the state image-like representation can be. Now if the map explodes to a 100x100, the transition states will always be of size v x v and training time will not exponentially explode.

Making these agents partially observable also greatly helped with the missed customer penalty mentioned in the earlier section. As the map becomes extremely large, there are dozens of customers that appear and disappear every timestep. Since we penalized every single agent in the system for missed customers, the signals became very convoluted and they were not able to recognize the importance of picking up customers. When we limited the state representation to just be of size v x v, the penalties attributed for missed customers also only applied to that partially observable view. Some of the initial concerns were that if an agent did not have a complete view of the map it may wander off in a undesirable location. Although, if enough iterations are done, each agent should have a good understanding of the map and where they are over a few timesteps. The drawback here is that the agents will not be able to generalize to other unseen maps that may have very different geographies.

## 4.4 TRANSFER LEARNING

The final method we will be talking about is the use of Transfer Learning to teach agents multiple objectives. In our AV fleet control problem, agents/cars have the main objective of fulfilling customer demand but have the secondary objective to charge its battery when needed. As state earlier

in the reward structure, agents received a positive reward when picking up customers but a large negative reward when it loses all of its energy. We found that when training on this reward structure, the agents did not seem to converge on any successful strategy. In some iterations, the agents only figured out how to go to the charging locations and stay there. In other cases, the agents just continued picking up customers until all the agents died. To solve this, we took advantage of transfer learning to consecutively teach these objectives. We begun with training the standard PO-MA-DRL model where the agents were given an infinite amount of energy. On the next iteration, the map and reward structure stayed consistent but this time we have each agent a random energy level from (50 minutes to 150 minutes). Now the agents were beginning to receive the large negative rewards but over time they were able to balance these two objectives and charge themselves when needed. In the second part of the Results/Tests we will show the results of this improvement.

## 5 EXPERIMENTS

In this section we provide two experimental results; first, we wanted to test the effectiveness of agent communication through divide and conquer techniques. Second, we tested the effectiveness of implementing transfer learning for teaching the agents to charge. In both tests, we compared the agents with baseline policies that we created. These baseline policies were build with Dijkstra's algorithm which calculates the shortest distance to each customer and chooses the action that will get to the customer the quickest. The goal of these experiments are to show the power of a DRL compared to a rules based Dijkstras approach.

### 5.1 EFFECTIVENESS OF RL MODEL VS. RULE BASED BASELINE

The goal of this evaluation is to simply test the ability of the agents to learning coordination policies as to maximize the reward of all the agents in the system. Our environment consisted of a 7x7 map with 2 agents similar to the example in figure 1. Our experiment randomly placed these agents on the board and randomized their priority. The map is in the form of a bridge in the middle and two opposite sides of the nap where customers appear randomly. The optimal cooperation policy here would be for the agents to split up and pick up customers on opposite sides of the map. We tested if our model can perform this divide and conquer coordination technique versus the baseline model which would just travel to the closest customer. Figure 2 shows the average customers fulfilled of the model versus the baseline.

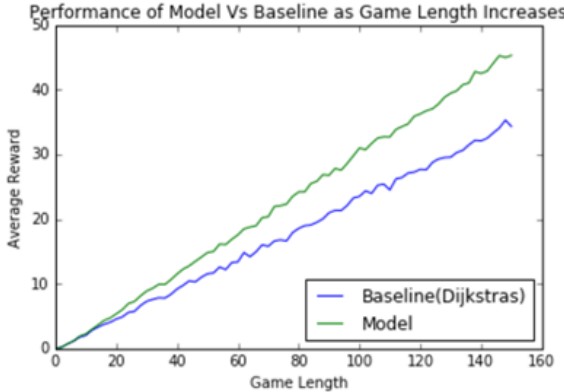

Figure 2: Customers Fulfilled vs. Game Length

We can see that as time (game length) increases the models reward begins to diverge from the baseline. This shows that the policies that the two model agents are performing more efficiently than the baselines. We inspected the individual games that were resulting and found that the agents did indeed perform a dived and conquer technique. When randomly placed on the same side of the map, one agent would take the initiative and travel to the other side of the map. When both agents started on different sides, they would just stay there and pick up their respective customers. This

simple example, successfully demonstrates that an agent can create a policy based on the other agent in the environment and the map structure to improve its overall expected future reward.

## 5.2 TRANSFER LEARNING TO TEACH MULTIPLE OBJECTIVES

The goal of this experiment was to demonstrate the ability to teach multiple objectives to agents through transfer learning and show its strength over a rules based approach. Our environment was a 10x10 bridge map with 4 agents like the one shown in figure 1. As described in the transfer learning section above, we initially trained a model to pick up customers with infinite energy. We then took that same trained model, gave each agent a random amount of energy and activated the penalty for losing all of your energy. When fine tuning, we found that we needed to set the epsilon for the e-greedy exploration to .5. Which meant that it would take the best action 50 percent of the time and a random otherwise. We linearly decayed this value to .01 over 3000 iterations. We found this approach to be successful in teaching the agent to keep itself alive and still pick up customers versus training a model altogether to learn to manage both objectives from scratch.

In our experiment we wanted to demonstrate that the RL model was successful in learning how to balance these two objectives effectively. Would the model stray too closely to keeping itself alive and only pick up a few customers? Or would it pick up many customers and sparingly recharge itself. We compared our model to two baseline agents. The first baseline agent would be a conservative agent where it would go back to the charging station if its energy was below 40 percent. The second, was a more aggressive baseline agent and would only go charge its energy when it was below 10 percent. Rule based simulation models would have to set these thresholds according to business requirements and goals. Setting a hard threshold is not the most optimal though as in some scenarios it may be more beneficial to be more aggressive and get the customers near you. In others, it helps to be more conservative as customers may be far away. The following table shows the performance of our model and both baseline agents on this map.

| Metric | Conservative Baseline | Aggressive Baseline | RL Model |
|:---:|:---:|:---:|:---:|
| Customers Fulfilled | 37.15 | 43.54 | **52.91** |
| Customers Missed | 16.95 | 14.7 | **12.71** |
| Net Reward | 20.2 | 28.84 | **40.19** |
| Agent Deaths | 0.0 | 11.76 | **2.31** |

We can see that the conservative baseline model successfully goes and charges it's battery when low but ends up picking up less customers than the aggressive baseline. Unfortunately, the aggressive baseline loses its energy more frequently and has experience around 11 deaths over the course of the trials. Our RL model shows that it is able to balance the deaths and customers fulfilled metrics and have an overall significant improvement in net reward. While it still may die from time to time, the overall performance boost of balancing these two objectives far exceeds the general baseline models.

## 6 CONCLUSION AND FUTURE WORK

Deep Reinforcement Learning provides a great approach to teach agents how to solve complex problems that us as humans may never be able to solve. For instance, Deep Mind has been successful in teach an agent to defeat the world champion in Go. More specifically, multi-Agent Reinforcement Learning problems provide an interesting avenue to investigate agent to agent communication and decision protocols. Since agents must rationalize about the intentions of other agents the dimensionality of the problem space becomes difficult to solve. In our use case, we wanted to see if we can scale a DRL solution up to an actual ride sharing environment that maintains the same dynamics as it would in real life. For this to be possible, we were tasked with the problem of teaching these agents effective cooperation strategies that would optimize the reward of the system along with the problem of teaching these same agents multiple objectives. This work, demonstrated how we successfully applied a partially observable multi-agent deep reinforcement solution to this ride sharing problem. Along with that, we showed how we can effectively take advantage of transfer learning to adapt decision policies to account for multiple objectives.

For future work, we plan to explore training individual networks per agents. This can be interesting to see the different strategies that agents learn respective of each other. From that, we can then test competition strategies and employ a sort of game theory problem. Just as how there are competing interest in the ride-sharing community between Uber and Lyft, we feel that this can be an interesting competition environment to learn from. Another area of future work can be to test and compare the effectiveness of other deep reinforcement learning technique like A3C or Actor Critic.

@incollectionBengio+chapter2007, author = Bengio, Yoshua and LeCun, Yann, booktitle = Large Scale Kernel Machines, publisher = MIT Press, title = Scaling Learning Algorithms Towards AI, year = 2007

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
