# OpenReview forum: "Autonomous Vehicle Fleet Coordination With Deep Reinforcement Learning"
_ICLR.cc/2018/Conference — Reject_

### Official Review · AnonReviewer1 · 2017-11-23
**no top-tier conference paper**

**Rating:** 3
**Confidence:** 5

**Review:**



This paper proposes to use deep reinforcement learning to solve a multiagent coordination task. In particular, the paper introduces a benchmark domain to model fleet coordination problems as might be encountered in taxi companies.

The paper does not really introduce new methods, and as such, this paper should be seen more as an application paper. I think that such a paper could have merits if it would really push the boundary of the feasible, but I do not think that is really the case with this paper: the task still seems quite simplistic, and the empirical evaluation is not convincing (limited analysis, weak baselines). As such, I do not really see any real grounds for acceptance.

Finally, there are also many other weaknesses. The paper is quite poorly written in places, has poor formatting (citations are incorrect and half a bibtex entry is inlined), and is highly inadequate in its treatment of related work. For instance, there are many related papers on:

-taxi fleet management (e.g., work by Pradeep Varakantham)

-coordination in multi-robot systems for spatially distributed tasks (e.g., Gerkey and much work since)

-scaling up multiagent reinforcement learning and multiagent MDPs (Guestrin et al 2002, Kok & Vlassis 2006, etc.)

-dealing with partial observability (work on decentralized POMDPs by Peshkin et al, 2000, Bernstein, Amato, etc.)

-multiagent deep RL has been very active last 1-2 years. E.g., see other papers by Foerster, Sukhbataar, Omidshafiei


Overall, I see this as a paper which with improvements could make a nice workshop contribution, but not as a paper to be published at a top-tier venue.

---

### Official Review · AnonReviewer3 · 2017-11-26
**Interesting problem and approach, but not ready for ICLR**

**Rating:** 3
**Confidence:** 3

**Review:**

In this paper, the authors define a simulated, multi-agent “taxi pickup” task in a GridWorld environment. In the task, there are multiple taxi agents that a model must learn to control. “Customers” randomly appear throughout the task and the taxi agents receive reward for moving to the same square as a customer. Since there are multiple customer and taxi agents, there is a multi-agent coordination problem. Further, the taxi agents have “batteries”, which starts at a positive number, ticks down by one on each time step and a large negative reward is given if this number reaches zero. The battery can be “recharged” by moving to a “charge” tile.

Cooperative multi-agent problem solving is an important problem in machine learning, artificial intelligence, and cognitive science. This paper defines and examines an interesting cooperative problem: Assignment and control of agents to move to certain squares under “physical” constraints. The authors propose a centralized solution to the problem by adapting the Deep Q-learning Network model. I do not know whether using a centralized network where each agent has a window of observations is a novel algorithm. The manuscript itself makes it difficult to assess (more on this later). If it were novel, it would be an incremental development. They assess their solution quantitatively, demonstrating their model performs better than first, a simple heuristic model (I believe de-centralized Dijkstra’s for each agent, but there is not enough description in the manuscript to know for sure), and then, two other baselines that I could not figure out from the manuscript (I believe it was Dijkstra’s with two added rules for when to recharge).

Although the manuscript has many positive aspects to it, I do not believe it should be accepted for the following reasons. First, the manuscript is poorly written, to the point where it has inhibited my ability to assess it. Second, given its contribution, the manuscript is better suited for a conference specific to multi-agent decision-making. There are a few reasons for this. 1) I was not convinced that deep Q-learning was necessary to solve this problem. The manuscript would be much stronger if the authors compared their method to a more sophisticated baseline, for example having each agent be a simple Q-learner with no centralization or “deepness”. This would solve another issue, which is the weakness of their baseline measure. There are many multi-agent techniques that can be applied to the problem that would have served as a better baseline. 2) Although the problem itself is interesting, it is a bit too applied and specific to the particular task they studied than is appropriate for a conference with as broad interests as ICLR. It also is a bit simplistic (I had expected the agents to at least need to learn to move the customer to some square rather than get reward and move to the next job from just getting to the customer’s square). Can you apply this method to other multi-agent problems? How would it compare to other methods on those problems?

I encourage the authors to develop the problem and method further, as well as the analysis and evaluation.

---

### Official Review · AnonReviewer2 · 2017-11-28
**The authors apply a previous algorithm named MADQN to the fleet management problem. Simulation results are not convincing and I have some questions regarding the partial observability.**

**Rating:** 4
**Confidence:** 4

**Review:**

The main contribution of the paper seems to be the application to this problem, plus minor algorithmic/problem-setting contributions that consist in considering partial observability and to balance multiple objectives. On one hand, fleet management is an interesting and important problem. On the other hand, although the experiments are well designed and illustrative, the approach is only tested in a small 7x7 grid and 2 agents and in a 10x10 grid with 4 agents. In spirit, these simulations are similar to those in the original paper by M. Egorov. Since the main contribution is to use an existing algorithm to tackle a practical application, it would be more interesting to tweak the approach until it is able to tackle a more realistic scenario (mainly larger scale, but also more realistic dynamics with traffic models, real data, etc.).

Simulation results compare MADQN with Dijkstra's algorithm as a baseline, which offers a myopic solution where each agent picks up the closest customer. Again, since the main contribution is to solve a specific problem, it would be worthy to compare with a more extensive benchmark, including state of the art algorithms used for this problem (e.g., heuristics and metaheuristics).

The paper is clear and well written. There are several minor typos and formatting errors (e.g., at the end of Sec. 3.3, the authors mention Figure 3, which seems to be missing, also references [Egorov, Maxim] and [Palmer, Gregory] are bad formatted).


-- Comments and questions to the authors:

1. In the introduction, please, could you add references to what is called "traditional solutions"?

2. Regarding the partial observability, each agent knows the location of all agents, including itself, and the location of all obstacles and charging locations; but it only knows the location of customers that are in its vision range. This assumption seems reasonable if a central station broadcasts all agents' positions and customers are only allowed to stop vehicles in the street, without ever contacting the central station; otherwise if agents order vehicles in advance (e.g., by calling or using an app) the central station should be able to communicate customers locations too. On the other hand, if no communication with the central station is allowed, then positions of other agents may be also partial observable. In other words, the proposed partial observability assumption requires some further motivation. Moreover, in Sec. 4.3, it is said that agents can see around them +10 spaces away; however, experiments are run in 7x7 and 10x10 grid worlds, meaning that the agents are able to observe the grid completely.

3. The fact that partial observability helped to alleviate the credit-assignment noise caused by the missing customer penalty might be an artefact of the setting. For instance, since the reward has been designed arbitrarily, it could have been defined as giving a penalty for those missing customers that are at some distance of an agent.

4. Please, could you explain the last sentence of Sec. 4.3 that says "The drawback here is that the agents will not be able to generalize to other unseen maps that may have very different geographies." In particular, how is this sentence related to partial observability?

---

### Decision · Program_Chairs · 2018-01-29
**ICLR 2018 Conference Acceptance Decision**

**Decision:**

Reject

**Comment:**

The reviewers agree that the manuscript is below the acceptance threshold at ICLR.  Many points of criticism were evident in the reviewer comments, including small artificial test domain, no new methods introduced, poor writing in some places, and dubious need for DeepRL in this domain.  The reviews mentioned a number of constructive comments to improve the paper, and we hope this will provide useful guidance for the authors to rewrite and resubmit to a future venue.